# Prevalence of Physical Activity and Sedentary Behavior among Chinese Children and Adolescents: Variations, Gaps, and Recommendations

**DOI:** 10.3390/ijerph17093066

**Published:** 2020-04-28

**Authors:** Hejun Shen, Jin Yan, Jin-Tao Hong, Cain Clark, Xian-Nan Yang, Yang Liu, Si-Tong Chen

**Affiliations:** 1School of Physical Education and Humanity, Nanjing Sport Institute, Nanjing 210014, China; 2Faculty of Education and Arts, University of Newcastle, Callaghan, NSW 2308, Australia; 3School of Physical Education and Sport Training, Shanghai University of Sport, Shanghai 200438, China; 4Centre for Sport, Exercise, and Life Sciences, Coventry University, Coventry CV1 5FB, UK; 5School of Physical Education and Sport, Beijing Normal University, Beijing 100875, China; 6Shanghai Research Center for Physical Fitness and Health of Children and Adolescents, Shanghai University of Sport, Shanghai 200438, China; 7Department of Physical Education, Shenzhen University, Shenzhen 518061, China

**Keywords:** moderate-to-vigorous physical activity, sedentary behavior, national monitoring and surveillance, research gaps, recommendations

## Abstract

Physical-activity (PA) and sedentary-behavior (SB) assessment is of particular importance in the promotion of health in young people. However, there is no comprehensive overview of PA and SB from national surveys among Chinese children and adolescents. Following a literature search for Chinese national health surveys, 11 papers from six national surveys were found. Of the included studies, the majority applied self-reported questionnaires to estimate the prevalence of PA and SB. Owing to different definitions of the prevalence of PA and SB and various measures, a large variation in prevalence of PA and SB was observed. Such variations were attributable to methodological and practical issues. This study highlights the current gaps in estimating the national prevalence of PA and SB among Chinese children and adolescents, which should be addressed. To improve the quality of PA and SB surveillance, standardized measurement protocols to estimate the prevalence of PA and SB more accurately among Chinese children and adolescents are urgently required.

## 1. Introduction

Regular engagement in moderate-to-vigorous physical activity (MVPA) and limiting sedentary behavior (SB) are considered independent risk factors of health outcomes in children and youth across the world [1,2,3,4]; additional beneficial health outcomes can be attained from concurrently accumulating MVPA and limiting SB [5,6,7,8,9]. There is substantive evidence to support the development of PA and SB guidelines, where it is recommended that young people should amass at least one hour of MVPA daily while limiting SB to under 2 h per day [10,11]. Many countries/nations and international organizations have established monitoring and surveillance systems to capture and understand the prevalence of PA and SB in children and youth, which, in turn, can be used to design effective interventions and inform policies for an active lifestyle. Hence, it is of vital importance to measure and assess the prevalence of PA and SB among young people [12].

Accurately capturing the prevalence of PA and SB is imperative and a prerequisite for future public-health actions and strategies. At present, the global prevalence of PA and SB in children and youth is at a concerning level. For example, the 2012 Lancet Physical Activity Series reported that nearly 80% of adolescents (13–15 years old) failed to meet PA guidelines, and over 60% of children had more than 2 h of daily SB [12]. As a global monitoring and surveillance tool with widespread impact, the Active Healthy Kids Report Cards in 2016 and 2018 informed that children and youth exhibited a lower level of PA and a higher level of SB [13,14]. Recently, Guthold and colleagues [15] found that 81% of 11–17-year-old students globally did not accrue sufficient PA. Insufficient PA and excessive SB are prevalent among Chinese children and adolescents. For example, two national surveys estimated that the prevalence of having at least 60 min of MVPA ranged from 29.9% to 34.1% [16,17], while the proportion of Chinese children and adolescents reporting no more than 2 h of SB was 65.4% [16]. However, inconsistency in reporting the prevalence of PA and SB among young people was observed [18]. For example, Pedisic et al. [19] indicated a large variation in the estimated prevalence of PA in Australian children and youth across different national surveillance programs. This inconsistency was largely attributed to measurement differences that, in turn, contribute to unreliable public-health insights.

Indeed, there exists a similar scenario in Chinese children and youth. Using the same dataset (2017 Physical Activity and Fitness in China—The Youth Study), Zhu et al. [16], and Liu et al. [20] estimated the prevalence of having at least 60 min of MVPA per day; however, Zhu et al. [16], and Liu et al. [20] reported that 34.1% and 13.1% of Chinese children and youth met the recommendation (60 min of MVPA), respectively. The noticeable variation of these two studies may be due to different measurement items for data collection. For example, Zhu et al. [16] estimated prevalence using the International Physical Activity Questionnaire (IPAQ), while Liu et al. [20] estimated it using Health Behavior in School-Aged Children (HBSC) questionnaire. Similar summaries were also derived from studies on SB prevalence [16,20,21]. The documented data suggest a large variation in PA and SB prevalence in monitoring and surveillance programs in China.

Thus, this study aimed to illustrate how differences of PA and SB measures influence prevalence in Chinese children and youth on the basis of results from multiple national surveillance systems. This study also sought to conduct critical analysis of the available literature, and then propose recommendations for future research and practices.

## 2. Method

### 2.1. Literature Search and Study Selection

To achieve the research aims of this study, we adopted a unique literature-search protocol. First, to the authors’ knowledge, only 9 national monitoring and surveillance programs involved in PA behavior have been conducted in China: (1) Chinese Youth Risk Behavior Survey (CYRBS) [22]; (2) Physical Activity and Fitness in China—The Youth Study (PAFCTYS) [23]; (3) China Health and Nutrition Survey (CHNS) [24]; (4) Global School-Based Student Health Survey (GSHS) [25]; (5) National Physical Fitness and Health Surveillance (NPFHS) [26]; (6) Chinese Children Dietary Index (CCDI) [27]; (7) China Education Panel Survey Junior High Cohorts study (CEPS) [28]; (8) China’s Report Card [20,29]; and (9) Chinese City Children and Youth Physical Activity Study (CCCYPA) [30]. After confirming the projects, we used 4 electronic databases, namely, Web of Science, EBSCO, ScienceDirect, and PubMed, to search for published articles derived from those monitoring and surveillance programs. We searched for articles of which the title or abstract included either the full surveillance programs title or abbreviations. Figure 1 presents the flowchart of study selection of this systematic review according to the Preferred Reporting Items for Systematic Reviews and Meta-Analyses (PRISMA) statement. Following this search protocol, a total of 2349 references were retrieved: 1280 from EBSCO, 191 from PubMed, 163 from ScienceDirect, and 760 from Web of Science. After removing duplicates, 966 references remained. Of all the remaining references, we screened on the basis of the inclusion and exclusion criteria (Table 1). Finally, 11 papers from 6 systems (GSHS, CHNS, CYRBS, PAFCTYS, China’s Report Card, and CCCYPA) were included [16,17,20,21,22,31,32,33,34,35,36]. The reference lists of 11 papers were also screened, which aimed to seek more eligible papers, but no more qualified papers were obtained. The literature search and study selection were independently performed by two authors (J.-T.H and X.-N.Y), and any inconsistency was resolved by a third author (S.-T.C).

### 2.2. Data Extraction

Information was extracted from the included studies regarding the first author, published year, project, study design, sample characteristics, PA and SB measures, definition of PA and SB prevalence, and PA and SB prevalence (%). Two authors independently extracted the information (J.-T.H and X.-N. Y), and any inconsistency between them was discussed with and resolved by a third author (S.-T.C) until consensus was achieved. If some studies reported the prevalence of PA and SB by age (grade) groups, sex, or other sociodemographic factors, those results were also extracted. The derived information from the included studies is presented in tabular format.

### 2.3. Quality-Assessment Study

Criteria for assessing the quality of studies were adapted from the Strengthening the Reporting of Observational studies in Epidemiology (STROBE) statement [37] and evaluation of the quality of prognosis studies in systematic reviews (QUIPS) [38]. On the basis of a previous review [39], for the PA and SB survey study, six items were regarded as the most important in the context of this review and included in the checklist: (1) study participation (whether reported in detail); (2) response rate (>80%); (3) objective measures (such measures as accelerometers); (4) validated measures (if study used subjective measures like self-reported questionnaires, reliability and validity had to be reported, or if a subjective measure was validated well, such as the HBSC questionnaire); (5) weighted prevalence estimates with 95% CI; and (6) weighted prevalence estimates stratified by sex, age, or other sociodemographic attributes. Score was assigned to each study on the basis of whether the quality-assessment items met the criteria (score = 1) or not (score = 0). The summed scores of each study ranged from 0–6, of which 0–2 was rated as low quality, 3–4 was rated as moderate quality, and 5–6 was rated as high quality. Two authors independently performed the quality assessment (J.-T H and X.-N. Y), and discrepancies between them was resolved by a third author (S.-T. C).

## 3. Results

Table 2 details all relevant information of the included studies. Specifically, these are 11 papers from six national surveillance projects (GSHS, CHNS, CYRBS, PAFCTYS, China’s Report Card, and CCCYPAS) that were published from 2007 to 2019. Of the 11 included papers, the single cross-sectional study design was the most common, while only two papers utilized a multiple cross-sectional design to capture trends of PA or SB prevalence. Only one study used objective measures (accelerometer) to monitor levels of PA and SB among Chinese children and adolescents, while most studies utilized self-reported questionnaires. Regarding the definitions of the prevalence of PA and SB, 60 min of moderate-to-vigorous PA (MVPA) per day, and no more than 2 h/day of SB were commonly used as cut-offs for determining the prevalence of PA and ST, respectively, in the included papers. The prevalence of PA and SB varied significantly across the included papers.

PA prevalence results by sex are presented in Figure 2, highlighting that, among all included studies, boys were more physically active that girls.

Of the five included studies on PA prevalence by age (Figure 3), three reported that older children and adolescents had lower PA prevalence.

SB prevalence by sex or age is presented in Figure 4 and Figure 5, respectively. However, according to the current summarized evidence, there was no clear sex- or age-specific SB prevalence.

Table 3 shows the results of quality assessment of each study included in this review. Of all included studies, quality scores ranged from 2 to 5. Overall, only three studies (out of 11) were rated as high-quality, while three and five studies were rated as low- and moderate-quality, respectively. 

## 4. Discussion

This review aimed to critically analyze national PA and SB surveys in Chinese children and adolescents, and identify gaps across those studies for future recommendations. This review found that only 11 national surveys investigated the prevalence of PA and SB among Chinese children and adolescents, with a range of strengths and inherent limitations, which must be addressed for future research and practices.

### 4.1. Key Findings and Interpretations

The most important research finding of this review is that there is a large variation in the reported PA and SB prevalence among Chinese children and adolescents across the previous studies. This finding is supported by previous studies [19]. Indeed, the main reasons for variations in the prevalence of PA and SB across different studies center at methodological differences, including study samples, used measures, and cut-offs for the definitions of PA and SB prevalence. First, owing to varying geographic and economic characteristics, Chinese children and adolescents in various regions have different lifestyles. Hence, different surveys consisting of different samples would necessarily lead to a large variation in the prevalence of PA and SB. Second, different measures for estimating the prevalence of PA and SB contribute to the variation. For instance, in our review, a study by Chen et al. [22] used a cut-off of five days of MVPA to measure PA prevalence, while Tian et al. [31] adopted a cut-off of seven days of MVPA, thus leading to inconsistency (19.9% vs. 12.2%). One interesting observation is that, despite the same sample data being analyzed in different studies, a large variation in prevalence was reported. Both Fan et al. [17], and Liu et al. [20] used the same sample but generated results with a large discrepancy. In estimating the prevalence of PA, Fan et al. [17] reported that 29.9% of Chinese children and adolescents met MVPA guidelines of 60 min per day, but Liu et al. [20] reported that the prevalence was 13.1%. The large variation between studies was likely attributable to the use of different measurement tools, where Fan et al. [17] used the IPAQ short form, while Liu et al. [20] used the HBSC questionnaire to collect PA data. The IPAQ was developed to monitor PA and SB among populations aged over 15 years old. In two studies by Fan et al. [17], and Zhu et al. [16], respectively, IPAQ was used to quantify time spent in MVPA that, in turn, was used to estimate PA prevalence among Chinese children and adolescents. Owing to the cognitive ability of younger children and adolescents (under 15 years old), recall bias is likely. The IPAQ was validated in a Chinese population aged over 15 years old [40,41]; thus, using the IPAQ to assess PA and SB levels among children and adolescents under 15 years old may be contentious and is likely contributing to inconsistent reporting in the literature.

Another issue related to assessing the prevalence of PA and SB among children and adolescents is self-reported measurement. Our review revealed that most national PA and SB monitoring and surveillance programs adopted self-reported questionnaires for estimation. Self-reported questionnaires appear to be the only viable instrument to estimate the prevalence of PA and SB because of lower cost effectiveness and testing burden; indeed, in the included papers of this review, the HBSC questionnaire developed by the WHO, and an interview-administered questionnaire developed by CNHS were the two most common measures to assess PA and SB prevalence among Chinese children and adolescents. However, their respective testing psychometrics should be considered. With regard to the HBSC questionnaire, its reliability was assessed, and it had satisfactory acceptability among urban adolescents in Beijing, China [42]. Such acceptable performance may not be replicated in other populations owing to different characteristics and is yet to be evaluated [43]. This concern should also be considered for validity and precision of the questionnaire [43]. To our knowledge, there is no evidence to demonstrate HBSC validity in assessing PA and SB among Chinese children and adolescents. In terms of interview-administered questionnaire-used studies by the China Health and Nutrition Study, there was no convincing evidence concerning its validity and reliability. Collectively, the prevalence of PA and ST estimated by self-reported measures among Chinese children and adolescents should be interpreted with extreme caution.

The majority of the included studies in our study adopted well-recognized cut-offs (PA: 60 min of MVPA per day; SB: no more than 2 h per day) [10] to assess whether Chinese children and adolescents were physically active and excessively sedentary, respectively. Such results and evidence provide comparable data for other types of research based on different populations. Although most studies in this review used internationally accepted cut-offs, different measurement approaches should be considered. Only one study, by Wang et al. [33], used objective measures (accelerometer) to discern levels of PA and SB in Chinese children and adolescents, while the remaining studies used subjective measures (including single or two items measures, or interview questionnaire). It is commonly believed that objective measures can be used to estimate PA and SB more accurately (intensity and duration) [44], as compared to subjective measures. It is, therefore, reasonable to assume that the estimated PA and SB prevalence among Chinese children and adolescents is inaccurate and fails to reflect true levels. Thus, future monitoring and surveillance programs should utilize more accurate measures to estimate the prevalence of PA and SB.

Our review highlighted that there are limited data on trends of PA and ST prevalence in Chinese children and adolescents. Indeed, PA and SB trends among children and adolescents are essential to understand health-behavior trajectories, improve future research, and inform public health policy. Since China’s Economic Reform and “opening-up” in 1978, rapid development in urbanization and industrialization occurred over the past four decades. Such advances contributed to more prevalent motorized lifestyles, characterized by insufficient PA and excessive SB among Chinese populations. In Chinese children and adolescents, because of heavy academic loads and pressure, available time for PA is reduced. Furthermore, with societal changes and development, PA and SB prevalence among Chinese children and adolescents are likely to be influenced. However, longitudinal PA and SB data are required to gain further insight into the importance of active lifestyles and their potential impact on population-health implications.

Using established cut-offs to determine PA and SB prevalence among Chinese children and adolescent, this study suggests that older children and adolescents have lower levels of PA, which is consistent with many previous studies [12,13,45]. In addition, boys are more physically active than girls, which is supported by other studies [12,13,45]. However, this review was unable to summarize SB patterns by sex and age among Chinese children and adolescents, indicating that more national studies on the prevalence of SB are needed.

### 4.2. Identified Gaps and Proposed Recommendations 

With the aforementioned key findings, this review also identified some gaps in previous surveys that must be mentioned and addressed.

#### 4.2.1. Accurate Prevalence of PA and SB

Owing to the current monitoring and surveillance practices of PA and SB among Chinese children and adolescents, an accurate picture of PA and SB prevalence is difficult to ascertain. To some extent, this situation limits researchers and policymakers to know the actual situation of active lifestyles among Chinese children and adolescents. Current researchers in China are more likely to estimate PA and SB prevalence among Chinese children and adolescents through self-reported measures, which may fail to provide an accurate prevalence of PA and SB, particularly relative to more objective assessment methods. To ensure the accurate assessment of PA and SB prevalence, key elements of data collection and processing protocols, including measures, survey administrations, survey time frames, and definitions of sufficient PA and limited SB should be standardized throughout all PA surveillance systems and population surveys in China.

In addition to estimating an overall prevalence of PA and SB, adjusting for various sociodemographic variables is necessary. The majority of current surveys only reported prevalence by sex and age. China is a country with complex social, economic, and geographic disparities, including multiple races, various societal stratifications, and noticeable differences in income and climate. We thereby advocate that future research should estimate the prevalence of PA and SB in subpopulations among Chinese children and adolescents.

#### 4.2.2. PA and SB Prevalence Trends 

A lack of studies on trends of the prevalence of PA and SB among Chinese children and adolescents is evident. This gap represents a barrier to longitudinally understanding trajectories of PA and SB prevalence among children and adolescents. Without trends of the prevalence of PA and SB, future interventional research and relevant policy designs are impractical and aimless. Researchers and/or governments should routinely representatively national measure PA and SB, which makes a systematic common surveillance system for PA and SB imperative of a health-monitoring system. Caution should be taken to reduce variability in sampling and testing procedures for actual trends of PA and SB to be attained. To better understand underlying causal mechanisms, trends should be adjusted for known factors to influence PA and SB among children and adolescents. It is documented well that PA and SB are determinants of population health, and should thus be examined concomitant to trends in other health indicators among children and adolescents. 

#### 4.2.3. PA and SB Measures

Although measures to assess PA and SB prevalence of the studies included in this review employ internationally recognized (e.g., HBSC or IPAQ) self-reported questionnaires, there is limited evidence regarding the validation of such measures for PA and SB among Chinese children and adolescents. Thus, more studies are urgently needed to examine the validity and reliability of self-reported measures for PA and SB among different subpopulations of Chinese children and adolescents.

Another important recommendation concerns the objective measurement for PA and SB among children and adolescents. Unlike studies of Western populations, research using objective measures to assess PA and SB among Chinese children and adolescents is understudied, with no available methodological consensus. For example, the predictive validity of laboratory-based prediction equations vs. count cut-offs in field settings is not known. It is, therefore, strongly recommended that researchers in China make concerted efforts to improve the quality of PA and SB assessment among children and adolescents.

### 4.3. Review Limitations

Despite some novelties of this review, some limitations should be mentioned. One major limitation is that only articles published in English were included in our review. Articles published in Chinese might give more information on this study topic. However, for non-Chinese speakers, the data of papers published in Chinese are not accessible for verification or reference. Another limitation is that few studies included in this review were rated as high-quality, while the majority were rated as low- or moderate-quality. However, to our knowledge, there has not been a recognized method to assess the quality of PA or SB survey studies. Future similar reviews should enhance the comprehensiveness of included studies, using an improved quality-assessment method to rate PA or SB survey studies among different populations. 

## 5. Conclusions

This study provides a comprehensive review and critical assessment of national surveys of PA and SB among Chinese children and adolescents. Currently, reported levels of PA and SB among Chinese children and adolescents may not be accurate due to some methodological and practical issues. There is urgent need for improved national PA and SB surveys among Chinese children and adolescents. In addition, Chinese researchers are encouraged to continue to study PA and SB measurement and assessment issues by conducting a greater number of evidence-based studies.

## Figures and Tables

**Figure 1 ijerph-17-03066-f001:**
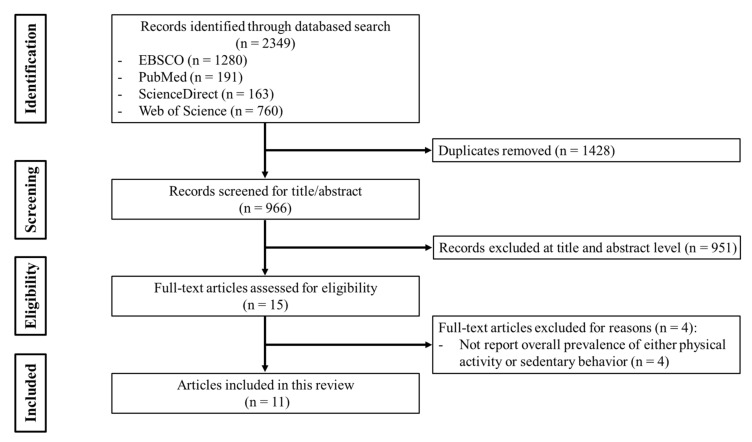
PRISMA flow diagram of study selection.

**Figure 2 ijerph-17-03066-f002:**
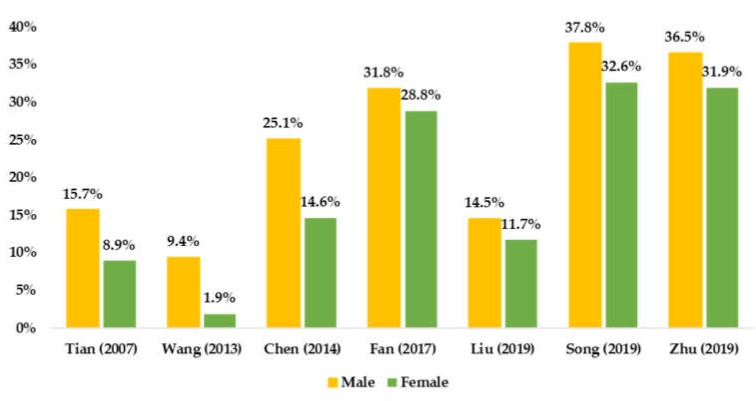
Summarized available prevalence of PA by sex among Chinese children and adolescents.

**Figure 3 ijerph-17-03066-f003:**
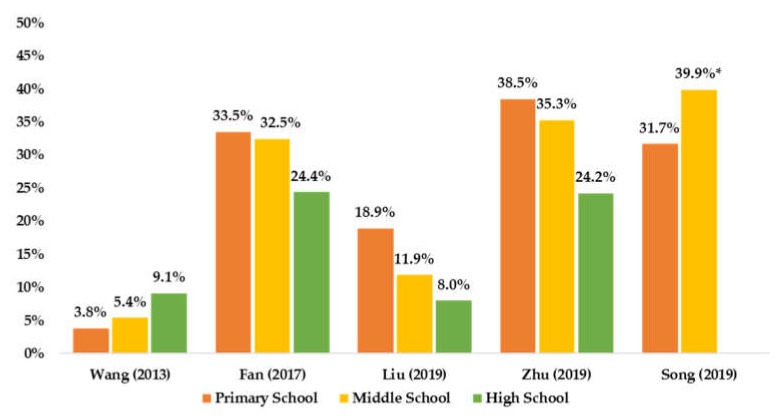
Summarized available PA prevalence by age group among Chinese children and adolescents. * combined middle school and high school.

**Figure 4 ijerph-17-03066-f004:**
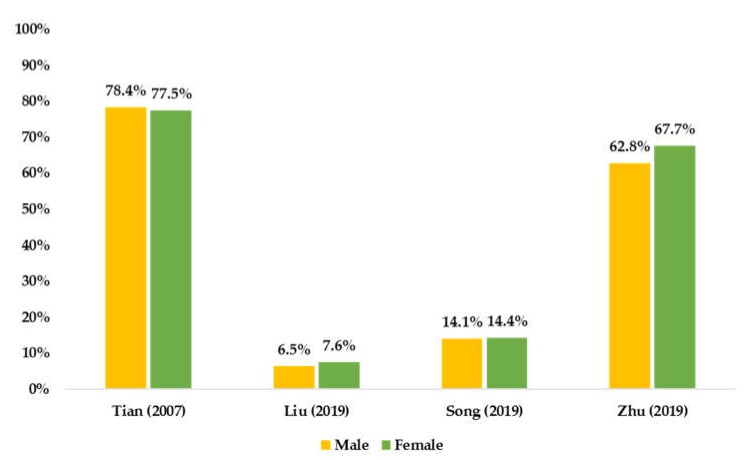
Summarized available prevalence of SB by sex among Chinese children and adolescents.

**Figure 5 ijerph-17-03066-f005:**
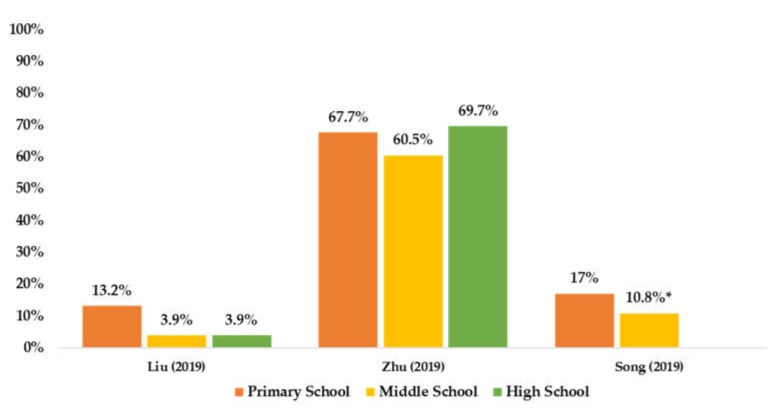
Summarized available prevalence of SB by age groups among Chinese children and adolescents.

**Table 1 ijerph-17-03066-t001:** Study inclusion and exclusion criteria.

Inclusion Criteria	Exclusion Criteria
Participants aged 6–19 years or in Grades 1–12; participants that were healthy or typically developed; studies that reported or included generally nationally representative samples, or included participants who were distributed to varied geographies and socioeconomic status; studies that reported prevalence of PA, prevalence of SB, or both; and published articles in referred academic journals.	Studies that: only reported results of participants in single or very few sites (less than three sites); included unhealthy or atypically developed participants; included participants over 19 years of age, above Grade 12, or below Grade 1; and were reviews or meta-analyses.

**Table 2 ijerph-17-03066-t002:** Review-study information.

Author (Year)	Design	Project	Sample	Measures		Prevalence Definition		Prevalence	
				PA	SB	PA	SB	PA (%)	SB (%)
Tian et al. (2007)	CS	China GSHS 2003	N = 7393 3633 m; 3760 f 13–15 years	GSHS		60 min MVPA/day	<3 h/day	O: 12.2; M: 15.7; F: 8.9	O: 78.0; M: 78.4; F: 77.5
Parvanta et al. (2010)	CS	CHNS 2004	N = 1552 828 m; 724 f 6–17.99 years	NM	Interview-administered questionnaire	NA	Low: <6 h; medium: 6–10.99 h; high: ≥11 h	NA	6–11.99 years Low: 23.3; medium: 38.8; high: 37.8. 12–17.99 years Low: 38.5; medium: 29.5; high: 32.0
Cui et al. (2011)	Multiple CS	CHNS 1997, 2000, 2004, 2006	N = 6817 3615 m; 3202 f 6–18 years	NM	Interview-administered questionnaire	NA	<2 h/day	NA	6–12 years *Urban* M: 1997: 86.7; 2000: 83.6; 2004: 68.3; 2006: 63.3 F: 1997: 86.5; 2000: 88.7; 2004: 66.3; 2006: 72.8 *Rural* M: 1997: 90.6; 2000: 88.5; 2004: 69.7; 2006: 53.3 F: 1997: 89.8; 2000: 88.4; 2004: 71.6; 2006: 57.7 13–18 years *Urban* M: 1997: 88.8; 2000: 88.9; 2004: 57.1; 2006: 55.7 F: 1997: 90.3; 2000: 91.5; 2004: 74.7; 2006: 65.3 *Rural* M: 1997: 94.3; 2000: 92.4; 2004: 77.1; 2006: 67.5 F: 1997: 97.0; 2000: 91.7; 2004: 83.3; 2006: 75.4
Zhang et al. (2012)	Multiple CS	CHNS 1997, 2000, 2004	N = 5497 2913 m; 2584 f 6–18 years	Interview-administered questionnaire		NC	≤2 h/day	6–11 years M: 1997: 71.5; 2000: 72.0; 2004: 72.4 F: 1997: 63.9; 2000: 65.9; 2004: 70.1 12–18 years M: 1997: 81.3; 2000: 84.0; 2004: 89.1 F: 1997: 79.0; 2000: 79.6; 2004: 80.6	O: 1997: 94.6; 2000: 93.5; 2004: 76.9 6–11 years O: 1997: 94.2; 2000: 93.1; 2004: 75.1 M: 1997: 94.9; 2000: 92.5: 2004: 74.5 F: 1997: 93.4; 2000: 93.8; 2004: 75.9 12–18 years O: 1997: 95.1; 2000: 93.8; 2004: 78.5 M: 1997: 94.5; 2000: 94.6; 2004: 75.6 F: 1997: 95.8; 2000: 92.8; 2004: 81.8
Wang et al. (2013)	CS	CCCYPAS	N = 2163 1086 m; 1077 f 9–17 years	GT3X accelerometers (MPA as 2800 to 3999 CPM and VPA as ≥4000 CPM;)		60 min MVPA/day	NA	O: 5.6; M: 9.4; F: 1.9 4–6th graders O: 3.8; M: 7.6; F: 1.0 7–9th graders O: 5.4; M: 8.4; F: 2.1 10–12th graders O: 9.1; M: 13.9; F: 3.1	NA
Chen et al. (2014)	CS	CYRBS 2011	N = 9901 5057 m; 4844 f 11–18 years	YRBSS	YRBSS	60 min MVPA/day≥ 5 days	None; >0–2 h; >2–4 h; >4 h	O: 19.9; M: 25.1;F: 14.6	*TV viewing* None (O: 57.0; M: 52.1; F: 62.2); >0–2 h (O: 38.2; M: 42.4; F: 33.9); >2–4 h(O: 3.5; M: 4.0; F: 3.0); >4 h (O: 1.2; M: 1.5; F: 0.9) *Computer use* None (O: 69.8; M: 66.0; F: 73.7); >0–2 h (O: 24.8; M: 26.8; F: 22.7); >2–4 h(O: 3.7; M: 4.7; F: 2.6); >4 h(O: 1.7; M: 2.5; F: 0.9)
Cai et al. (2017)	CS	PAFCTYS 2016	N = 116,615 57,685 m; 58,930 f 9–17 years	NM	Chinese Version of HBSC	NA	≤2 h/day	NA	Primary *Urban* O: 66.8; M: 64.8; F: 68.9 *Rural* O: 65.0; M: 59.5; F: 70.3 Junior middle *Urban* O: 59.5; M: 57.4; F: 61.6 *Rural* O: 60.5; M: 55.5; F: 65.5 Junior high *Urban* O: 61.8; M: 60.7; F: 63.0 *Rural* O: 69.0; M: 68.0; F: 69.9
Fan and Cao (2017)	CS	PAFCTYS 2016	N = 90,712 42,644 m; 48,068 f 9–17 years	IPAQ—Short Form	NM	60 min MVPA/day	NA	O: 29.9; M: 31.8; F: 28.2 *Urban* M: 33.6; F: 29.0 *Rural* M: 30.6; F:27.9 Primary O: 33.5; M: 34.1; F: 33.0 *Urban* M: 36.5; F: 36.5 *Rural* M: 32.1; F: 29.4 Junior middle O: 32.5; M: 34.1; F: 31.2 *Urban* M: 37.3; F: 32.4 *Rural* M: 31.6; F: 30.6 Junior high O: 24.4; M: 27.8; F: 21.3 *Urban* M: 27.7; F: 18.6 *Rural* M: 28.5; F: 24.1	NA
Liu et al. (2019)	CS	China Report Card 2018	N = 125,281 62,139 m; 63,142 f 9–17 years	Chinese Version of HBSC		60 min MVPA/day	<2 h/day	O: 13.1; M: 14.5; F: 11.7 Primary 18.9; secondary 11.9; upper secondary 8.0 Rural: 12.6; Urban: 13.8	O: 7.1; M:6.5; F: 7.6 Primary 13.2; secondary 3.9; upper secondary 3.9
Song et al. (2019)	CS	CHNS 2010-2012	N = 38,744 19,631 m; 19,113 f 6–17 years	Interview-administered questionnaire		NC	<2 h/day	O: 35.4; M: 37.8;F: 32.6 6–12 years 31.7; 13–17 years 39.9	O: 14.2; M: 14.1;F: 14.4 6–12 years 17.0; 13–17 years 10.8
Zhu et al. (2019)	CS	PAFCTYS 2017	N = 105,246 50,557 m; 54,689 f 7–19 years	IPAQ—Short Form	Chinese Version of HBSC	60 min MVPA/day	≤2 h/day	O: 34.1; M: 36.5; F: 31.9 Primary 38.5; Junior middle 35.3; Junior high 24.20 Urban: 34.1; Rural: 34.1	O: 65.4; M: 62.8; F: 67.7 Primary 67.7; junior middle 60.5; junior high 69.70 Urban: 66.6; Rural: 64.0

Note: PA = physical activity; SB = sedentary-behavior time; CS = cross-sectional; M/m = males; F/f = female; h = hours; O = overall; MVPA = moderate-to-vigorous physical activity; MPA = moderate physical activity; VPA = vigorous physical activity; NM = not measured; NA = not available; NC = not clear; GSHS = Global Student Health Survey; CHNS = China Health and Nutrition Survey; YRBSS = the Youth Risk Behavior Surveillance System; PAFCTYS = Physical Activity and Fitness in China—The Youth Study; IPAQ = International Physical Activity Questionnaire; CCCYPAS = Chinese City Children and Youth Physical Activity Study.

**Table 3 ijerph-17-03066-t003:** Quality assessment of each study included in this review.

Author and Year	Study Participation		Measurements		Data Analysis		Total	Quality Judgement
	Describe Participants in Detail	Response Rate >80%	Objective Measures	Validated Measures	Prevalence (95% CI)	Stratification		
Tian et al. (2007)	1	1	0	1	1	1	5	High
Parvanta et al. (2010)	1	0	0	0	0	1	2	Low
Cui et al.(2011)	1	0	0	0	0	1	2	Low
Zhang et al. (2012)	1	0	0	0	0	1	2	Low
Wang et al. (2013)	1	0	1	1	0	1	4	Moderate
Chen et al. (2014)	1	1	0	1	0	1	4	Moderate
Cai et al.(2017)	1	1	0	1	1	1	5	High
Fan and Cao (2017)	1	1	0	1	1	1	5	High
Liu et al.(2019)	1	1	0	1	0	1	4	Moderate
Song et al. (2019)	1	0	0	0	1	1	4	Moderate
Zhu et al. (2019)	1	1	0	1	1	1	5	Moderate

0, not met criteria; 1, met criteria; 0–2, low quality; 3–4, moderate quality; 5–6, high quality.

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
