# Peer review of "Prevalence of Physical Activity and Sedentary Behavior among Chinese Children and Adolescents: Variations, Gaps, and Recommendations"

_ijerph, 2020, doi:10.3390/ijerph17093066_

Round 1
Reviewer 1 Report
This study aimed to show the prevalence of PS and ST in Chinese children and youth, to make a critical analysis of literature and to suggest some improvements for future research.
The study is well aligned. However, there are a few aspects that need to be corrected:
- ST is sometimes mentioned as Screen or Sedentary time. They are related but they are not exactly the same.
- In line 77, Methods Section, you mention that you have only “7 monitoring surveillance programs involved..” but you mention 9.
- Line 154: complete citation.
- In line 210, Discussion section, you mention that boys are more PA than girls. However, figure 3 dues not match this statement.
- The references section must be improved.
Best regards,
Reviewer 2 Report
Thank you for the opportunity to review the manuscript titled “Prevalence of Physical Activity and Sedentary Time among Chinese Children and Adolescents: Variations, Gaps, and Recommendation”. This review summarises and discusses the body of literature on physical activity and sedentary time in Chinese children. The manuscript is well written and presented. I only offer a few minor comments to assist the authors in strengthening their manuscript.
- P2L86: Remove the redundant word ‘that’ before the word ‘whose’.
- P2L87: Why limit the search to titles and abstracts? Given many journals’ word constraints for titles and abstracts, some papers may not have been able to provide the full name of the surveillance programs in their title or abstract. It may therefore have been useful to also search full manuscript texts.
- P2L89: Spell out Web of Science instead of using the abbreviation.
- P3L96: What about studies that included data from participants aged 6-19 years as well as younger and/or older children and young persons? Were such studies excluded even if they reported separate data for eligible and ineligible study populations?
- P9L123: Consider removing the overall prevalence from Figure 1 to more cleanly show differences by sex. The overall prevalence is sufficiently reported in Table 2 and the manuscript text.
- P9L123: Ditto comment above. Consider removing the overall prevalence from Figure 3 to more cleanly show differences by sex. The overall prevalence is sufficiently reported in Table 2 and the manuscript text.
- Although the authors discuss some of the limitations in the body of literature, including methodological issues and measures of PA and ST, there is inadequate description and discussion of the limitations of this review. I recommend that the authors add a paragraph describing the limitations of their review and discuss the implications of these limitation, especially in regard to interpretation of the findings.
Reviewer 3 Report
Very nice paper about how differences of physical activity and sedentary time measures influence the prevalence in Chinese children and youth, based on the results from multiple national surveillance systems. This study also sought to conduct a critical analysis of the available literature, and then propose some recommendations for future research and practice. This is an important question that remains to be clarified but this work is a step forward in the knowledge about this subject. There are some specific changes and suggestions that should be made to improve the quality of the paper.

Round 2
Reviewer 1 Report
Dear Editor,
the authos have amended all the suggested changes.
Best regards,